PSENEN influences the progression of renal clear cell carcinoma by regulating the immune microenvironment and oxidative phosphorylation

Huang Congying 1 2
Chen Kaijie 1 2
Zhu Siyu 1
Yang Xin 3
Hou Jiangang 4 Hou_jiangang@126.com
Gu Xuefeng 1 2 guxf@sumhs.edu.cn
1 School of Pharmacy, Shanghai University of Medicine & Health Sciences , Shanghai, Shanghai , China
2 School of Health Sciences and Engineering, University of Shanghai for Science and Technology , Shanghai, Shanghai , China
3 Department of Endocrinology, Putuo People’s Hospital, Tongji University , Shanghai, Shanghai , China
4 Department of Urology, Huashan Hospital, Fudan University , Shanghai, Shanghai , China
Uversky Vladimir
Electronic publication date: 2024 Nov 29
Publication date: 2024
Volume: 12
Electronic Location ID: e18457
Received 2024 Jul 23; Accepted 2024 Oct 14
Copyright: © 2024 Huang et al.
Copyright year: 2024
Copyright holder: Huang et al.
License: This is an open access article distributed under the terms of the Creative Commons Attribution License, which permits unrestricted use, distribution, reproduction and adaptation in any medium and for any purpose provided that it is properly attributed. For attribution, the original author(s), title, publication source (PeerJ) and either DOI or URL of the article must be cited.
License URL: https://creativecommons.org/licenses/by/4.0/

Keywords: PSENEN (PEN2), Metformin, Kidney renal clear cell carcinoma, Prognosis, Immune microenvironment, Oxidative phosphorylation

Funding: Shanghai University of Medicine & Health Sciences E3-0200-21-201011-87 Cultivation Plan for Improving the Quality of Degree Programs in Colleges and Universities A1-2601-24-203001 Teacher Professional Development Project-Industry-University-Research Practice A3-0200-24-311008-17 The Construction Project of Shanghai Key Laboratory of Molecular Imaging 18DZ2260400 The study was supported by a university-level research fund from Shanghai University of Medicine & Health Sciences (E3-0200-21-201011-87), the Cultivation Plan for Improving the Quality of Degree Programs in Colleges and Universities (A1-2601-24-203001), and the Teacher Professional Development Project-Industry-University-Research Practice (A3-0200-24-311008-17). The Construction Project of Shanghai Key Laboratory of Molecular Imaging (18DZ2260400) supported the APC of this article. The funders had no role in study design, data collection and analysis, decision to publish, or preparation of the manuscript.

==============================
Background

Presenilin enhancer gamma-secretase subunit (PSENEN), the straight target of metformin, is highly expressed in several cancers. The role of PSENEN in kidney renal clear cell carcinoma (KIRC) has not been reported.

Methods

PSENEN expression in KIRC specimens was investigated in The Cancer Genome Atlas (TCGA) and Gene Expression Omnibus (GEO) databases, as well as by immunohistochemical analysis and qPCR assay. The relationship between PSENEN expression and patient survival was discussed. The biological function of PSENEN in KIRC and its correlation with immune infiltration of KIRC were then investigated, and possible cellular mechanisms were again analyzed. The effects of metformin on KIRC cell proliferation, migration and invasion were discussed in cellular experiments.

Results

PSENEN was found to be highly expressed in KIRC. The high PSENEN expression was an adverse factor in KIRC. Several immune-related pathways were enriched including immune response, complement and coagulation cascade reactions, and neutrophil extracellular trap formation, as evidenced by enrichment analyses. Immune infiltration analysis revealed that PSENEN expression correlated positively with regulatory T cells. Gene set variation analysis suggested that PSENEN expression correlated positively with oxidative phosphorylation. In addition, a certain concentration of metformin was found to inhibit the proliferation, migration and invasion of KIRC cells, in which PSENEN down-regulation, AMPK up-regulation and mTOR down-regulation were also observed.

Conclusions

PSENEN may be involved in regulating the immune microenvironment of KIRC, and oxidative phosphorylation may also be a pathway for its involvement in cancer development. PSENEN is a novel prognostic marker for KIRC.

Introduction

Metformin is usually used to treat hyperglycemia in patients with type 2 diabetes (Ma et al., 2022). Although the role of metformin in various diseases has been widely reported over the years, the presenilin enhancer gamma-secretase subunit (PSENEN) has only recently been identified as its direct molecular target (Ma et al., 2022). Metformin inhibits glucose production in the liver by activating AMP-activated protein kinase (AMPK), thus exerting a hypoglycemic effect (Ravindran et al., 2017). Metformin plays a role in anti-aging. For instance, vascular smooth muscle cell senescence causes vascular aging, and metformin inhibits vascular smooth muscle cell aging by enhancing autophagy (Tai et al., 2022). Several scholars have demonstrated a good antitumor effect of metformin in some cancers (e.g., breast cancer, blood cancer, colorectal cancer, endometrial cancer) (Feng, Jia & Shen, 2022; Han et al., 2023; Lan et al., 2022; Liu et al., 2019). Specifically, the activation of AMPK by metformin and the inhibition of PKM2 under glucose deprivation conditions yields better therapeutic outcomes in patients with renal cell carcinoma (Liu et al., 2019).

Presenilin enhancer gamma-secretase subunit (PSENEN) forms the protein complex γ-secretase with presenilin, nicastrin, and anterior pharynx defective 1 (APH1). γ-secretase can cleave proteins in transmembrane domains (Klein et al., 2022). PSENEN is a small membrane protein required to promote proteolysis in presenilin (Klein et al., 2022). PSENEN binds to presenilin to regulate γ-secretase activity and participate in amyloid beta-peptide (Abeta) production (Cai & Tomita, 2020; Mao et al., 2012). After beta-amyloid precursor protein is cleaved by γ-secretase, it releases Abeta, the accumulation of which in the brain is thought to be a major cause of Alzheimer’s disease (Mao et al., 2012; Wolfe, 2020). Some related studies suggested that mutations in PSENEN might lead to Alzheimer’s disease (Klein et al., 2022). Additionally, some researchers have found that mutations in PSENEN are inextricably linked to the development of hidradenitis suppurative/acne inversa and Dowling-Degos disease (Xiao et al., 2020; Ralser et al., 2017; Pavlovsky et al., 2018). PSENEN has also been linked to the development of neurodevelopmental disease. Specifically, the PSENEN-Notch signaling pathway plays an essential role in maintaining neural stem cells during cortical development (Cheng et al., 2019). Previous studies show significantly upregulated levels of PSENEN expression in most human cancers compared to normal tissues (Chen et al., 2022). Surprisingly, PSENEN is associated with survival and immune infiltration in low-grade gliomas (Chen et al., 2022). However, the role of PSENEN is less studied in kidney cancer (KC).

The incidence of KC has remained high (Sung et al., 2021). Kidney renal clear cell carcinoma (KIRC) is one of the most prevalent subtypes of KC, accounting for more than 80% of cases of renal malignancy (Hsieh et al., 2017; Brozovich et al., 2021). Although immunotherapy and targeted therapy have been applied in the treatment, most patients do not achieve remission (Rao Ullur et al., 2023). Therefore, there is a pressing need to validate useful biomarkers or methods for clinical treatment and prognostic diagnosis for better therapeutic outcomes. This study investigated the expression and prognostic value of PSENEN in KIRC. Multiple datasets and immunohistochemistry of clinical samples were first used to analyze PSENEN expression in KIRC. Second, the relevance of the association of PSENEN expression with the survival of KIRC patients was discussed by plotting survival curves. Gene Ontology (GO) functional analysis and Kyoto Encyclopedia of Genes and Genomes (KEGG) analysis were performed to explore the biological function of PSENEN and investigate its relevance with the immune microenvironment of patients with KIRC. Finally, possible cellular mechanisms were further explored by Gene Set Variation Analysis (GSVA). Our cellular experiments clarified the effect of metformin on KIRC progression. Based on our results, we hypothesize that PSENEN is a survival-related gene.

Materials and Methods

Databases and samples

The transcriptome data and clinical data of 784 samples were collected in this study. The specific information is summarized in Table 1. The RNA sequencing data of 599 sample RNA sequencing data samples were downloaded from the Cancer Genome Atlas (TCGA, https://cancergenome.nih.gov/) database. Gene expression profiles of the GSE150404 dataset were obtained from Gene Expression Omnibus (GEO, https://www.ncbi.nlm.nih.gov/geo/) database (Hutter & Zenklusen, 2018; Barrett et al., 2011). A total of 73 normal tissue samples were obtained from the Genotype-Tissue Expression (GTEx, https://www.gtexportal.org) database (Aran et al., 2017). Moreover, 23 pairs and five pairs of clinical kidney cancer samples were collected for immunohistochemistry and qPCR assays. Tumor samples were provided by Huashan Hospital of Fudan University (Shanghai, China). This study was approved by the Ethics Committee of Huashan Hospital of Fudan University (KY2011-009). Written informed consent was obtained from the study participants.

Table 1 Clinical characteristics of kidney renal clear cell carcinoma cohorts.

Clinical characteristic	TCGA KIRC	GSE150404	GTEx	Immunohistochemistry	qPCR	
(n = 599)	(n = 60)	(n = 73)	(n = 42)	(n = 10)	
Type						
Tumor	527	60	–	23	5	
Normal	72	–	73	23	5	
Grade						
G1	14	–	–	6	–	
G2	224	–	–	10	2	
G3	206	–	–	5	3	
G4	75	–	–	2	–	
Stage						
I	264	15	–	–	–	
II	56	15	–	–	–	
III	122	15	–	–	–	
IV	82	15	–	–	–	

Semi-quantitative immunohistochemistry analysis

The tissue samples were fixed with formaldehyde and embedded in paraffin wax, and then sliced by microtome (Ma et al., 2021). The slices were immersed in xylene I and xylene II (Titan, Shanghai, China) for 10 min each, and in 100%, 95%, 90%, 80%, 70%, and 50% anhydrous ethanol (Titan, Shanghai, China) for 5 min each. This was subsequently repeated thrice in distilled water for 5 min, for dewaxing and hydration. Next, antigen repair was induced by heat at 120 °C for 10 min and cooled to indoor temperature. All slides were covered with the diluted PSENEN primary antibody (Bio-Techne, Minneapolis, MN, USA) and incubated overnight at 4 °C, followed by incubation with the horseradish peroxidase labeled PSENEN secondary antibody (Absin, Shanghai, China) for 1 h at the indoor temperature. Color development was achieved by staining with DAB (Yeasen, Shanghai, China) for 5 min and re-staining with hematoxylin (Servicebio, Wuhan, China) for 1 min. Successive slides of the same renal carcinoma tissue were stained and observed in the same field of view. The positive expression of PSENEN was localized to the cytoplasm of cancer cells. PSENEN expression and those of other proteins were individually assessed as follows. A: cell staining intensity was calculated as 0 (negative staining); 1 (weakly positive staining); 2 (moderately positive staining), and 3 (strongly positive staining). B: Area stained was assessed as follows: 0 (0–5%); 1 (6–25%); 2 (26–50%); 3 (51–75%), and 4 (>75%). C: PSENEN protein expression in each section was determined by multiplying the scores of A and B: score of 1–4 was defined as weakly positive (+); 5–8 as moderately positive (++), and 9–12 as strongly positive (+++). Immunohistochemical experimental results were analyzed using SPSS 20.0 software. Count data were expressed as proportions (%). The X2 test was used to statistically analyze the correlation between the semi-quantitative expression of PSENEN in each group of samples and the clinical and pathological characteristics of patients. Considering the items with frequencies less than 5, Fisher’s exact test was used for verification, and p < 0.05 was considered as a statistically significant difference.

RNA isolation and real-time quantitative polymerase chain reaction

Total RNA was extracted from tissues and cells by TRIzol reagent (Invitrogen, ThermoFisherScientific, Shanghai, China) following the manufacturer’s protocol. The extracted RNA was reverse transcribed into cDNA using PrimeScript™ RT reagent Kit (TaKaRa Bio, Beijing, China) (Tang et al., 2019). TB Greene Premix Ex Tag™ (Tli RNase Plus, TaKaRa Bio, China) was used to determine the levels of mRNA expression. The expression of target RNAs was normalized to that of glyceraldehyde-3 phosphate dehydrogenase (GAPDH) based on the 2−ΔΔCt method (Tang et al., 2019). Specific primer information for RT-qPCR is presented in Table 2. The results of real-time quantitative polymerase chain reaction (RT-qPCR) experiments were used to compare the differences between cancer tissues and adjacent tissues by paired t-test. Finally, the results were visualized through GraphPad Prism 9.5 software, and p < 0.05 was considered as a statistically significant difference.

Table 2 Primer sequences.

Primer	Sequences	
GAPDH-F	CAAGGCTGAGAACGGGAAG	
GAPDH-R	TGAAGACGCCAGTGGACTC	
PSENEN-F	GAAGTGAGCTCTCCTGGGTCAAG	
PSENEN-R	TTCTCCTCATTGGACACTCGC	
AMPK-F	CGGCAAAGTGAAGGTTGGC	
AMPK-R	CCTACCACATCAAGGCTCCG	
mTOR-F	GACCTCTTCTCCTTGGCACA	
mTOR-R	TCTGGTGTCAGGGTATCCCA	

Cell

KIRC cell line A498 cells were purchased from Coweldgen Scientific (Shanghai, China), and authenticated by short tandem repeat profiling analysis. A498 cells were cultured in Dulbecco’s modified Eagle’s medium (Gibco, Grand Island, NY, USA) supplemented with 10% fetal bovine serum (Gibco, Grand Island, NY, USA) and 1% penicillin-streptomycin (Gibco, Grand Island, NY, USA) in a humidified incubator at 37 °C with 5% CO2.

Cell proliferation assay

Cell proliferation assays were performed by the CCK-8 method. The KIRC cell line, A498, was inoculated in 96-well plates and incubated at 37 °C in an atmosphere with 5% CO2 for 24 h. Then metformin medium at 1, 5, 10, and 15 mM was added to the wells, and the wells without metformin medium were used as controls. After incubation in the incubator for 0, 24, 48, and 72 h, the CCK-8 reagent (APExBIO, Houston, TX, USA) was added and incubation was continued for 2 h. The absorbance at 450 nm was measured using a microplate reader (Thermo, Waltham, MA, USA, FisherScientific, Shanghai, China). GraphPad Prism 9.5 software was used for data analysis and to create a line graph. The F-test was used to compare the differences between samples, and p < 0.05 was considered as a statistically significant difference.

Wound healing assay

A498 cells were inoculated in six-well plates, and when the cells reached confluency, the wound was scraped using a 10 μl pipette tip, and then the wounded surface was washed thrice with PBS. Next, 1 mmol and 10 mmol of metformin media were added to the wells, and the wells without metformin medium served as controls. The movement of cells toward the wounded area was photographed at 0, 12 and 24 h. Image J was used to process the scratch pictures for each group at different times, set a scale, circle the scratch area, and calculate the scratch area and the migration rate. GraphPad Prism 9.5 software was used to perform a significant difference analysis on the results. The F-test was used to compare the differences between samples, and p < 0.05 was considered as a statistically significant difference.

Transwell assay

Complete culture medium containing different concentrations of metformin was firstly prepared and set aside. The density of A498 cells in serum-free medium was adjusted to 5 × 103 cells/200 μL, and the cells were seeded in Transwell cell culture chambers (Labselect, Hefei, Anhui, China). The lower chambers were loaded with medium containing different concentrations of metformin, and then cultured in a cell culture incubator for 48 h. The cells were washed twice with calcium-free PBS and fixed with 4% paraformaldehyde for 20 min, and then stained with 0.1% crystal violet for 10 min. The cells were washed twice with calcium-free PBS, fixed with 4% paraformaldehyde for 20 min, and stained with 0.1% crystal violet for 10 min. The upper layer of non-migrated cells was gently wiped away with a wet cotton ball. After rinsing with PBS, the cells were observed under a 10× microscope (Leica, Wetzlar, Germany), and four fields of view were selected for photographs, and the infiltrating cells were counted by Image J. GraphPad Prism 9.5 software was used to perform a significant difference analysis on the results. The F-test was used to compare the differences between samples, and p < 0.05 was considered as a statistically significant difference.

Differential gene expression analysis

To increase the sample size of normal tissue sequencing specimens for PSENEN differential expression analysis, information from TCGA and GTEx databases was integrated. All transcriptome data were in the log2 (TPM+1) format. Differences between groups were analyzed using the Wilcox test. Differences in PSENEN expression scores in 21 pairs of cancer and adjacent tissues were analyzed using the Student’s t-test.

Prognostic analysis

Whether PSENEN expression was an independent prognostic target in KIRC patients was examined using a univariate Cox regression model (Dai et al., 2021). Using Kaplan Meier-Plotter networking tools (http://kmplot.com/analysis/), the Kaplan Meier curve was plotted to assess the correlation between PSENEN expression and patient prognosis (Hou et al., 2017).

Enrichment analysis

TCGA-KIRC cohort tumor samples were divided into two groups of high and low expression based on the median PSENEN expression, using the edgeR package for differential analysis, and the cutoff threshold of |log2 (FC)|>1, p < 0.05. The obtained differentially expressed gene (DEG) sets were analyzed by GO functional analysis and KEGG pathway enrichment analysis to study the biological function of PSENEN expression in KIRC (Ebrahimie et al., 2017; Du et al., 2014).

CIBERSORT analysis

CIBERSORT analysis was used for immune filtration studies in TCGA dataset (Kawada et al., 2021). Then Pearson correlation analysis was used to determine the correlation between PSENEN expression and the abundance of 22 kinds of immune infiltrates from CIBERSORT.

Gene set variation analysis

Gene set variation analysis (GSVA) was performed using the R package “GSVA” to further explore the potential biological functions of PSENEN expression in KIRC (Zhang et al., 2022). The relevant genes were downloaded from the molecular signature database (MSigDB, https://www.gsea-msigdb.org/) for GSVA. p-value < 0.05 was considered statistically significant.

Results

PSENEN expression is elevated in KIRC

To investigate the effect of PSENEN in KIRC progression, we first analyzed its expression in normal and tumor samples using the TCGA-KIRC and GTEx datasets. Compared with normal samples, the expression level of PSENEN was increased in KIRC samples (Fig. 1A). Subsequently, it was also found in the TCGA-KIRC dataset that the expression level of PSENEN increased with the elevation of the World Health Organization (WHO) tumor grade (Fig. 1B), and the same trend was observed in the tumor node metastasis (TNM) stage (Fig. 1C). In addition, the expression of PSENEN was obviously elevated from stage I to III in the GSE150404 dataset (Fig. 1D). These results of bioinformatics analysis indicate that PSENEN is highly expressed during KIRC progression.

Figure 1 PSENEN expression in KIRC.

(A) PSENEN expression in KIRC was studied using the TCGA and GTEx datasets (tumor: n = 527, normal: n = 145). (B, C) The connection of PSENEN expression level with WHO grade (G1: n = 14, G2: n = 224, G3: n = 206, G4: n = 75) and TNM stage (I: n = 264, II: n = 56, III: n = 122, IV: n = 82) in the TCGA database. (D) The connection of PSENEN expression level with TNM stage in the GSE150404 database (I: n = 15, II: n = 15, III: n = 15, IV: n = 15). “*”, “**” and “***” indicate p < 0.05, p < 0.01 and p < 0.001, respectively.

Subsequently, we compared the tumor and paracancer tissues from KIRC specimens of four different tumor stages (Fig. 2A). Immunohistochemical analyses suggested that PSENEN expression levels in KIRC tumor tissues were markedly higher compared to that in the paracancer tissues (Fig. 2B), and its expression levels increased continuously with the deterioration of cancer (Fig. 2C). qPCR assays on five pairs of cancer and paracancer tissues yielded similar results, with PSENEN expression levels increasing as G2 progressed to G3. PSENEN expression was significantly upregulated in cancer tissues overall in all samples (Figs. 2D–2E). These results were consistent with those of bioinformatics analysis. In general, PSENEN expression is elevated in KIRC.

Figure 2 Immunohistochemical and qPCR results of tumor and paraneoplastic (normal) tissues from KIRC patients.

(A) The representative PSENEN immunohistochemical staining in KIRC tissues (G1: n = 6, G2: n = 10, G3: n = 5, G4: n = 2). (B) PSENEN expression in KIRC tissues and normal tissues (tumor: n = 23, normal: n = 23). (C) The connection of PSENEN expression level with WHO grade in KIRC tissues (G1: n = 6, G2: n = 10, G3: n = 5, G4: n = 2). (D, E) The results of qPCR assays for five groups of cancer and paracancerous tissues (tumor: n = 5, normal: n = 5; G2: n = 2, G3: n = 3). “*”, “**” and “***” indicate p < 0.05, p < 0.01 and p < 0.001, respectively.

Upregulated PSENEN expression is an adverse factor for KIRC

We evaluated the correlation between PSENEN expression levels and the prognosis of patients with renal clear cell carcinoma. Univariate Cox regression analysis suggested that the enhanced PSENEN expression was related to poor overall survival (OS), disease-specific survival (DSS), and progression-free interval (PFI) in patients (Fig. 3A). Subsequently, the results from the Kaplan-Meier survival analysis revealed that patients with high PSENEN expression had a worse prognosis (Fig. 3B), and shorter OS, DSS, and PFI, which was consistent with some of the previous analysis results of Yang et al. (2023) (Figs. 3C–3E). Among them, PFI takes the progression after disease treatment as the endpoint event. PFI is generally considered to be a better choice of clinical endpoint than OS and DSS, because patients usually experience disease recurrence before death, and thus more endpoint events are recorded during the follow-up period. In summary, high PSENEN expression is a potential adverse factor for KIRC.

Figure 3 Associations between PSENEN expression levels and the survival prognosis of patients with KIRC.

(A) Forest plot showing the hazard ratios of PSENEN in KIRC. (B) Prognostic value of PSENEN expression in KIRC patients based on the Kaplan Meier database. Kaplan Meier survival curves of (C) OS, (D) DSS, and (E) PFI for patients stratified according to PSENEN expression profiles in KIRC based on TCGA database.

PSENEN may potentially regulate the immune microenvironment of KIRC

We conducted GO and KEGG analyses to explore the biological function of PSENEN in KIRC (Figs. 4A, 4B). Interestingly, among the biological processes (BP) obtained from GO analysis, PSENEN was found to be associated with immune-related biological processes, such as humoral immune response, antimicrobial humoral response, complement activation, humoral immune response mediated by circulating immunoglobulin, and antimicrobial humoral immune response mediated by antimicrobial peptide. The cellular component (CC) terms mainly included immunoglobulin complexes, circulating immunoglobulin complexes, etc. Moreover, KEGG analysis revealed that the function of PSENEN was related to neuroactive ligand-receptor interactions, complement and coagulation cascades, and neutrophil extracellular trap formation. PSENEN may be involved in the regulation of inflammatory responses and immune microenvironment in patients with KIRC.

Figure 4 The results of pathway enrichment analyses, immune infiltration analysis and GSVA.

(A) Top 10 BP, CC and MF enrichment terms of DEGs. (B) Top 20 KEGG enrichment pathways of DEGs. The blue dots represent the q values of the upper axis enrichment pathways, and the horizontal bars correspond to the gene ratio in the lower axis enrichment terms. (C) The relevance between PSENEN and 22 types of infiltrating immune cells. (D) The results of GSVA and correlation analysis. “*”, “**” and “***” indicate p < 0.05, p < 0.01 and p < 0.001, respectively.

It is well known that immune infiltrating cells have a significant impact on cancer progression. We performed an immune infiltration analysis using the CIBERSORT algorithm (Fig. 4C). The analysis revealed that PSENEN expression was positively correlated with the infiltration level of six types of immune cells, including regulatory T cells, T helper cells, CD8 T cells, activated NK cells, macrophages, and gamma delta T cells. PSENEN expression was negatively correlated with the infiltration of five types of immune cells, namely macrophages M2, resting mast cells, natural killer cells, naïve B cells, and eosinophils. Thus, PSENEN may influence tumorigenesis and cancer progression by regulating the infiltration of multiple immune cells.

PSENEN may be involved in the development of KIRC through oxidative phosphorylation

To further explore the functional significance of PSENEN levels in KIRC, possible cellular mechanisms were analyzed by GSVA. KEGG gene sets, hallmark gene sets, and WikiPathways gene sets were downloaded for GSVA using the MSigDB database, and the results of relevancy analysis showed that PSENEN expression was positively related to oxidative phosphorylation (Fig. 4D). Among them, the positive correlation based on hallmark gene sets is relatively weak, which may be due to the differences in the scope of gene sets, because hallmark gene sets focus on core components, while KEGG and WikiPathways gene sets cover a broader network. Oxidative phosphorylation is one of the important pathways in the metabolic reprogramming of KIRC and is essential for the development of KIRC (Akhtar, Al-Bozom & Al Hussain, 2018). Therefore, PSENEN may affect cancer progression through oxidative phosphorylation.

Metformin inhibits the proliferation, migration and invasion of KIRC cells

Cell proliferation assay was performed using the KIRC cell line, A498, by adding four concentrations of metformin, 1, 5, 10, and 15 mM. The results showed that within 72 h of cell culture, the addition of metformin significantly inhibited the ability of cancer cells to proliferate compared to the control group, and the effect became more pronounced with increasing metformin concentrations (Figs. 5A, 5B). Metformin effectively slowed down the migration ability of cancer cells after 24 h of cell culture compared to the control group, as evidenced by the wound healing assay, and 10 mM of metformin inhibited cell migration more significantly (Figs. 5C, 5D). Subsequently, Transwell experiments were conducted and it was able to observe that metformin significantly inhibited the invasion of A498 in a dose-dependent manner compared to the control group (Figs. 5E, 5F). In addition, qPCR experiments showed that the expression level of PSENEN was significantly down-regulated in metformin-treated A498 cells (Fig. 5G). It was also found that the expression of AMPK mRNA was significantly up-regulated in metformin-treated A498 cells compared to the control group, as well as the expression of mTOR mRNA was significantly down-regulated in metformin-treated A498 cells, both in a dose-dependent manner (Figs. 5H, 5I). Therefore, it was hypothesised that activation of AMPK or inhibition of mTOR pathway may be one of the important mechanisms by which metformin affects the proliferation, migration and invasion of KIRC cells, and that the expression level of PSENEN was significantly down-regulated during the oncogenic effect of metformin.

Figure 5 The effect of metformin on proliferation, migration and invasion of KIRC cells.

(A, B) The cell proliferation assay showed the influence of different concentrations of metformin on the proliferation ability of A498 cells (in each group, n = 3). (C, D) The wound healing assay showed the influence of different concentrations of metformin on the migration ability of A498 cells (in each group, n = 3). (E, F) The Transwell assay showed the influence of different concentrations of metformin on the invasive ability of A498 cells (in each group, n = 4). (G) PSENEN mRNA expression in A498 cells treated with different concentrations of metformin. (H) AMPK mRNA expression in A498 cells treated with different concentrations of metformin. (I) mTOR mRNA expression in A498 cells treated with different concentrations of metformin. “*”, “**”, “***” and “****” indicate p < 0.05, p < 0.01, p < 0.001 and p < 0.0001, respectively.

Discussion

Through continuous exploration by researchers, PSENEN has been confirmed as a direct target for metformin and acts as an anti-glycemic agent by activating the AMPK pathway (Ma et al., 2022). PSENEN is an important part of the composition of γ-secretase (Klein et al., 2022). Mutations in PSENEN adversely affect the development of several diseases, such as Alzheimer’s disease, hidradenitis suppurative/acne inversa, and Dowling-Degos disease (Xiao et al., 2020; Ralser et al., 2017; Pavlovsky et al., 2018). Additionally, PSENEN is not only involved in adipocyte differentiation but is also associated with neurodevelopmental diseases (Cheng et al., 2019). However, the role of PSENEN has not been widely explored in cancer.

KIRC is the subtype of KC that causes the most deaths (Hsieh et al., 2017). With the development and application of diagnostic and therapeutic approaches, such as immune checkpoint inhibitor therapy and anti-vascular endothelial growth factor therapy, the prognosis of some patients has substantially improved (Rao Ullur et al., 2023; Jaszai & Schmidt, 2019). However, tumor progression and possible drug resistance often lead to treatment failure (Jaszai & Schmidt, 2019; Hegde & Chen, 2020). Consequently, the identification of new biomarkers and therapeutic targets is urgent.

In this study, we first bioinformatically analyzed the data of PSENEN expression in the TCGA-KIRC dataset and the GEO database’s GSE150404 dataset. Next, we performed immunohistochemistry and qPCR assays using clinical samples. Our analysis uncovered that PSENEN was obviously upregulated during the development of KIRC. We then further investigated the prognostic value of PSENEN expression in KIRC using the Kaplan-Meier plotter. By survival analysis and univariate Cox regression analysis, we demonstrated that the elevated expression level of PSENEN in KIRC is an adverse factor.

To explore the molecular mechanisms of PSENEN action in KIRC, we conducted GO and KEGG analyses which uncovered that some enriched pathways were associated with inflammatory response and immune regulation, such as humoral immune response, immunoglobulin complexes, complement and coagulation cascade reactions, and neutrophil extracellular trap formation. The complement system is an essential element of the congenital immune response, and complement activation leads to inflammatory responses and the recruitment of immunoreactive cells (Bajic et al., 2015). Separately, neutrophils are among the inflammatory cells enriched in the tumor microenvironment (TME). Neutrophil extracellular traps (NETs) are reticular structures released by neutrophils to trap microorganisms and their main function is antimicrobial defense (Van Avondt & Hartl, 2018). NETs are involved in regulating the activity and metabolism of cancer cells (Wang et al., 2020; Yang et al., 2020; Berger-Achituv et al., 2013). These results suggest that there might be relevance of PSENEN expression for immune responses in patients with KIRC.

KIRC is a highly immune infiltrating tumor. The higher the percentage of regulatory T cells, the worse the outcome of KIRC (Vuong et al., 2019). The immune inhibitor molecule, CTLA4, is associated with poor prognosis of patients with KIRC (Zhang et al., 2019). CTLA4 is associated with regulatory T cells (Pedros et al., 2017). Notably, PSENEN expression was positively correlated with regulatory T cell infiltration levels in our immune infiltration results. Metformin exerts anticancer effects by modulating the immune microenvironment (Foretz, Guigas & Viollet, 2023; Wang et al., 2020). Specifically, metformin inhibits mTORC1 activation-induced differentiation of naïve CD4 T cells to inducible regulatory T cells by reducing the levels of Foxp3 protein, thereby reducing the abundance of tumor-infiltrating regulatory T cells, and also downregulated effector molecules such as CTLA4 and IL-10 (Kunisada et al., 2017). The inhibition of AMPK restores the production of inducible regulatory T cells, revealing the involvement of mTORC1 and AMPK (Kunisada et al., 2017). Metformin stimulates PSENEN to bind to the ATP6AP1 subunit and inhibits lysosomal proton pump v-ATPase activity, resulting in the activation of the AMPK pathway. The PSENEN-ATP6AP1 axis is necessary for the inhibition of mTORC1 signaling (Ma et al., 2022). In this study, the results of CCK-8 assay, wound healing assay and Transwell assay showed that metformin inhibited the proliferation, migration and invasion ability of KIRC cells. And down-regulation of PSENEN, up-regulation of AMPK and down-regulation of mTOR were observed in metformin-treated KIRC cells. Therefore, it is speculated that activation of AMPK or inhibition of the mTOR pathway may be one of the important mechanisms of action of metformin in affecting the proliferation, migration and invasion of KIRC cells, and that the level of PSENEN may be related to the immune microenvironment of cancer, but it remains to be explored in depth whether it is a direct target of the anticancer effect of metformin.

KC is also considered a metabolic disease, and the array of genes mutated in KC occurrence is involved in regulating various metabolic events such as glycolysis, tri-carboxylic acid (TCA) cycle, oxidative phosphorylation, and fatty acid metabolism (Chakraborty et al., 2021; Wettersten, 2020; Weiss, 2018). Oxidative phosphorylation is the main pathway that supports the production of sufficient ATP for cell growth (Huang et al., 2022). In KC, tumor cells preferentially derive energy through the aerobic glycolysis-mediated hypoxic response pathway rather than by oxidative phosphorylation (Akhtar, Al-Bozom & Al Hussain, 2018). As the activity of the TCA cycle decreases, oxidative phosphorylation activity is impaired in cancer cells (Fontana, Anselmi & Limonta, 2023). The study by Huang et al. (2022) identified a subset of cells in KIRC that predicted patient survival with high accuracy, had lower levels of oxidative phosphorylation and higher immune infiltration characteristics; overexpression of target genes activated oxidative phosphorylation and inhibited apoptosis. Activation of oxidative phosphorylation has a protective effect on KIRC (Huang et al., 2022). Further GSVA and correlation analyses in our study demonstrated that the expression levels of PSENEN correlated positively with oxidative phosphorylation. Therefore, oxidative phosphorylation may be a potential pathway for PSENEN action in influencing the development of KIRC.

Conclusions

In summary, our study reports for the first time, the role of PSENEN expression in KIRC and its association with the prognosis of patients with KIRC. Our study suggests that high PSENEN expression in KIRC is an adverse factor, which may be a direct or indirect participant in immune regulation in cancer progression and is closely associated with oxidative phosphorylation. PSENEN may be a new effective biomarker for targeting KIRC but its specific mechanism of action remains to be thoroughly validated.

Supplemental Information

Supplemental Information 1 Raw Data.

Additional Information and Declarations

Competing Interests

Author Contributions

Human Ethics

DNA Deposition

Data Availability

The authors declare that they have no competing interests.

Congying Huang analyzed the data, prepared figures and/or tables, authored or reviewed drafts of the article, and approved the final draft.

Kaijie Chen analyzed the data, prepared figures and/or tables, authored or reviewed drafts of the article, and approved the final draft.

Siyu Zhu performed the experiments, authored or reviewed drafts of the article, and approved the final draft.

Xin Yang performed the experiments, authored or reviewed drafts of the article, and approved the final draft.

Jiangang Hou conceived and designed the experiments, authored or reviewed drafts of the article, and approved the final draft.

Xuefeng Gu conceived and designed the experiments, authored or reviewed drafts of the article, and approved the final draft.

The following information was supplied relating to ethical approvals (i.e., approving body and any reference numbers):

This study was approved by the Ethics Committee of Huashan Hospital of Fudan University (KY2011-009).

The following information was supplied regarding the deposition of DNA sequences:

The RNA sequencing data of 599 sample RNA sequencing data samples were downloaded from the Cancer Genome Atlas (TCGA, https://cancergenome.nih.gov/) database. Gene expression profiles of the GSE150404 dataset were obtained from Gene Expression Omnibus (GEO, https://www.ncbi.nlm.nih.gov/geo/) database.

The following information was supplied regarding data availability:

NCBI GEO database: GSE150404.

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
