# Peer review of "PSENEN influences the progression of renal clear cell carcinoma by regulating the immune microenvironment and oxidative phosphorylation"

_PeerJ, doi:10.7717/peerj.18457_

## Round 0.1 · original submission · Major Revisions

Please address concerns of all reviewers and amend manuscript accordingly

Reviewer 1 ·

Basic reporting

The manuscript is clear in presenting the rational for the study within the introduction. In support of the authors premise, the background information and references presented links associated concepts of metformin, AMPK, and the immune response in relation to PSENEN in worsening KIRC.

However, the lack of detail and clarity throughout the study, unfortunately leads to a greater concern about novelty, rigor, and validity.

1. Previously published work (cited below) presenting a pan-cancer analysis already aims to highlight the role of PSENEN in the prognosis and immunology of cancer. With figures 3C and 3D in this study bearing similarity to figures 3A and 3B of the cited study, respectively. Could the authors please explain more clearly the novelty of this data presented in the context of the current manuscript and provide suitable referencing/citations for this? (Zerui, Yang & Wen, & Dingsheng, & E, Y & Yubing, & Kai, Chen & Zhikun, Qiu & Xingyun, Liu & Xiong, L. (2023). Pan-cancer analysis highlights the role of PSENEN in the prognosis and immunology of cancer. 10.1016/S2707-3688(23)00097)

2. In the same vein, Figure 3B bears a similarity to the Kaplan-Meier plot presented in the Human Protein Atlas for PSENEN in KIRC (https://www.proteinatlas.org/ENSG00000205155-PSENEN/pathology/renal+cancer/KIRC). The authors would need to explain more clearly the novelty of this data presented in the context of the current manuscript and provide suitable referencing/citations for this if applicable?

5. Images within figures 2 and figures 5 are of a poor quality/resolution and do not contain scale bars. Like wise all panels in figure 1 are differently formatted either in size of graph, style of graph, and/or clarity of image.

3. Figure legends - only give a very basic description of figure content. i.e., Figure 5 beings with " A, B The results of cell proliferation assay. C, D The results of wound healing assay. E The
results of Transwell assay"; which at no point gives any useful detail.

4. Added to this, figure legends are lacking in detail and occasionally incorrect. An example being Figure 5F to 5I being mislabeled and Figure 5I being absent from the legend.

6. Figures and graphs are missing appropriate detail and/or axis titles. I.e., Although it can be inferred the authors should highlight which of figures 1A and B is the WHO grade and which is the TNM Grade. Also what are the units for gene expression, is this fold-increase? Figures 2D/E say "Genen" is this correct?

7. Lines 191-197 are vague and would require being reformed to make the specific datasets being talked about more clear. With the paragraph stating the same fact four times without differentiating between the points well enough.

Experimental design

I am doubtful all methods as described would be repeatable based on the detail given within this study alone.

1. Please ensure reagents are appropriately listed and ideally manufacturer for all compounds used. I.e., no indication of which PSENEN antibody was used in IHC of figure 2.

2. Similarly there are gaps in explaining methodology. What software and criteria was used to assess cell movement in wound healing assay? It would be useful for the authors to share the R Code utilized in the study as supplementary.

3. My understanding of CIBERSORT analysis is the requirement of both an RNA data set being analyzed and a reference dataset. What is this reference data set? Does TGCA offer both aspects?

4. I presume Table 1 is an indication of n-numbers for each assay? If I am right this detail would be better placed with the given data (i.e., figure legend). The cell proliferation assay and migration rate has no n-numbers and the raw data just says average for each group. How many is in the group? and also at the 0h time point the OD Value average in the raw data is the same for all 5 conditions. Do these come from the same group or do all groups normally average at the same OD?

Validity of the findings

I can grasp the concept that the authors are trying to convey based on the described connection between Metformin - glucose - liver - AMPK - oxidative phosphorylation. However, I find little novelty in the relationship between PSENEN and KIRC (figure 1 & 3), which can be found in existing literature and databases. While the evidence for the parts which could be novel are weak and unfortunately present little connection to immune infiltration and/or oxidative phosphorylation that is convincing. Especially when the authors only provide a single figure of evidence for immune response (figure 4; which is only bioinformatic) and a single figure of evidence for metformin effects (figure 5). Which is not sufficient as metformin could have alternative effects that are not adequetly discussed in this paper.

A recommendation would be to enhance the clarity and novelty of figures 1-3 which would act to support a deeper story of PSENENs role in the immune response in association with metformin. A narrative which would require in vitro analysis of the immune response which could include flow cytometry and/or tumor IHC for immune response (i.e., CD8 for T-cells or CD206 for M2-like Macrophages; based on the CIBERSORT presented in this study).

Reviewer 2 ·

Basic reporting

Manuscript is clear and defined in the professional English language. The literature provides sufficient references, tables and figures in the context of the subject.

Experimental design

Research is well described with appropriate methods and sufficient detail.

Validity of the findings

Conclusions are well stated and linked to original research.

·

Basic reporting

The manuscript is written in clear, professional English, adhering to appropriate standards of technical and scientific communication. The background and introduction provide sufficient context and refer to relevant literature, demonstrating the study's fit within the broader field. The article is well-structured, following standard sections, and figures are relevant, properly labeled, and of high resolution. Raw data appears to be appropriately shared in accordance with data sharing policies. The study is self-contained and presents all relevant results to support its hypotheses without unnecessary subdivision of content.

Experimental design

The manuscript presents original research within the journal's scope, addressing a well-defined, relevant question about PSENEN's role in KIRC, filling a key knowledge gap. The investigation meets high technical and ethical standards, with methods described in sufficient detail for replication. More detailed statistical analysis descriptions could be provided to further enhance clarity.

Validity of the findings

The manuscript adheres to the journal's guidelines, with no subjective evaluation of impact or novelty. The research provides a meaningful contribution to the field, and all underlying data are robust, statistically sound, and well-controlled. The conclusions are well-stated, appropriately linked to the original research question, and supported by the presented results. Any claims made are aligned with the findings, avoiding overreach in interpretation.

Additional comments

The manuscript investigates the role of PSENEN in the progression of kidney renal clear cell carcinoma (KIRC), particularly its influence on the immune microenvironment and oxidative phosphorylation. The study's bioinformatic and experimental analyses provide strong evidence for PSENEN as a potential prognostic marker. Some minor clarifications, such as detailing statistical analyses and addressing potential confounders, would further strengthen the robustness of the conclusions. Please see below for more details.

Figure 1
(Minor point) Possible duplication of scatter plot dots in the box plot function (Figure 1A). This is likely a plotting artifact, where data points are visualized both as a jitter or strip plot and overlaid on a boxplot.

Figure 2(D/E)
With such a small sample size (n=2 for G2 and n=3 for G3), the statistical power to detect meaningful differences is very low. Non-parametric or exact tests could be used for the analysis, but the small sample sizes remain a significant challenge for reliable interpretation.
Additionally, the manuscript lacks an overall description of the statistical analyses used either in the methods or results sections. Figure 2B, for example, the data is ordinal, the use of non-parametric tests would be more appropriate than parametric tests like the t-test. The specific statistical tests used are not mentioned, which is critical for accurate interpretation. This omission should be corrected, also in sections where parametric tests (e.g., t-test or Wilcoxon) might have been applied.

Figure 3
1. The manuscript does not provide context for DFI, but it is included in Figure 3A. Please either remove DFI from Figure 3A or provide an explanation of DFI in the main text to justify its inclusion.
2. The analysis presented in Figure 3A is univariate. It does not account for potential confounders such as age, gender, tumor stage, and grade. It would be beneficial to perform multivariate Cox regression analysis to ensure that the observed associations between PSENEN expression and survival are independent of these confounding factors. Similar logic is also applicable to Figure 1A (also Table 1). Fig1A, for example, patient comorbidities (e.g., diabetes or hypertension) might influence gene regulatory pathways and alter PSENEN expression. Stratification by these factors would help reduce the risk of confounding if the data is available.

Figure 4
1. I agree with the conclusion that PSENEN expression is positively associated with oxidative phosphorylation, supported by strong correlations in the KEGG and WikiPathways gene sets. However, the weaker correlation with the Hallmark gene set should be briefly addressed. This may be due to differences in gene set scope, as Hallmark focuses on core components while KEGG and WikiPathways cover broader networks. A brief explanation would clarify this discrepancy and strengthen the interpretation.
2. It is not clear how the top 10 (GO) or top 20 (KEGG) pathways were selected.

Reviewer 4 ·

Basic reporting

1.I believe the legend needs updating, for instance: “Migration ability of A498 cells with different concentrations of metformin for 24h” may be more appropriate than “The results of XX assay.”
2.Drug concentration units in the article need to be standardized. For example, “10 mmol of metformin” in Line 252 should be corrected to 10mM or 10mmol/L.
3.Additionally, I suggest getting a native English speaker to polish the article.

Experimental design

Research indicate that a high oral dose of metformin is required to induce AMPK phosphorylation in mice, while in clinical patients, the concentration of metformin is insufficient to activate AMPK in cells. I suggest detecting the expression levels of PSENEN2 and p-MAPK proteins treated with metformin in the renal cancer cell line A498 with western blotting, and further validating the findings in several renal cancer cell lines or in vivo experiments.

Validity of the findings

no comment

Annotated reviews are not available for download in order to protect the identity of reviewers who chose to remain anonymous.

---

## Round 0.2 · accepted · Accept

All concerns of the reviewers were addressed and revised manuscript is acceptable now.

·

Basic reporting

no comment

Experimental design

no comment

Validity of the findings

no comment

Additional comments

My initial review concerns are all well addressed in this revised version.